# Effect of Refining and Fractionation Processes on Minor Components, Fatty Acids, Antioxidant and Antimicrobial Activities of Shea Butter

**DOI:** 10.3390/foods12081626

**Published:** 2023-04-12

**Authors:** Adel G. Abdel-Razek, Ghada A. Abo-Elwafa, Eman F. Al-Amrousi, Ahmed N. Badr, Minar Mahmoud M. Hassanein, Ying Qian, Aleksander Siger, Anna Grygier, Elżbieta Radziejewska-Kubzdela, Magdalena Rudzińska

**Affiliations:** 1Department of Fats and Oils, National Research Centre, Dokki, Cairo 12622, Egypt; 2Department of Food Toxicology and Contaminants, National Research Centre, Dokki, Cairo 12622, Egypt; 3Faculty of Food Science and Nutrition, Poznań University of Life Sciences, Wojska Polskiego 28, 60-637 Poznań, Poland

**Keywords:** shea butter, shea olein, shea stearin, refining, fractionation, sterols, fatty acids, tocopherols, antioxidant activities, antimicrobial activities

## Abstract

Shea butter is becoming increasingly popular in foods, cosmetics and pharmaceutical products. This work aims to study the effect of the refining process on the quality and stability of fractionated and mixed shea butters. Crude shea butter, refined shea stearin, olein and their mixture (1:1 *w*/*w*) were analyzed for fatty acids, triacylglycerol composition, peroxide value (PV), free fatty acids (FFA), phenolic (TPC), flavonoid (TFC), unsaponifiable matter (USM), tocopherol and phytosterol content. Additionally, the oxidative stability, radical scavenging activity (RSA), antibacterial and antifungal activities were evaluated. The two main fatty acids in the shea butter samples were stearic and oleic. The refined shea stearin showed lower PV, FFA, USM, TPC, TFC, RSA, tocopherol and sterol content than crude shea butter. A higher EC50 was observed, but antibacterial activity was much lower. The refined olein fraction was characterized by lower PV, FFA and TFC in comparison with crude shea butter, but USM, TPC, RSA, EC50, tocopherol and sterol content was unchanged. The antibacterial activity was higher, but the antifungal activity was lower than those of crude shea butter. When both fractions were mixed, their fatty acid and triacylglycerol composition were similar to those of crude shea butter, but other parameters were different.

## 1. Introduction

The exciting properties of shea butter could explain the wide range of its uses, especially in the food, cosmetics and pharmaceutical industries. Shea butter is taken out of its nut using different methods, such as boiling, semi-mechanical, mechanical and solvent extraction methods. Honfo et al. [1] showed that the quality of the extracted shea butter depended on where it came from, how it was treated before extraction and how it was extracted. Davrieux et al. [2] reported that a significant difference in the average fat content of shea kernels from different origins was noticed, where the fat content of East African shea kernels (52.92 g/100 g) was significantly higher than that of West African (48.03 g/100 g). Maanikuu and Peker [3] reported that the composition of shea butter includes a saponifiable triglyceride fraction composed mainly of stearic, oleic acids and lower amounts of palmitic, linoleic and arachidic acids. It also contains a high percentage of unsaponifiable fractions (triterpenes, tocopherols, phenols and sterols), which are bioactive substances having anti-inflammatory and antioxidant properties that give shea butter its importance and medicinal properties. It acts as an emollient and skin moisturizer and has sun-screening properties. It also has anti-aging properties, and its consumption reduces blood cholesterol and protein concentrations in serum and organs. However, the food industry utilized only the hard fraction (shea stearin) in past years. Lovett [4] mentioned that 80% of all annual shea exports were unprocessed shea kernels. It is well-known that shea butter refining involves various steps. When making refined shea butter from raw shea butter, there are four important steps: de-gumming, neutralization, bleaching and deodorization. The ingredients that are not good for consumption are removed from the butter at each stage of the refining process. Additionally, the refining process unintentionally removes some desirable natural bioactive components. After being refined, shea butter can be fractionated into shea butter olein and stearin, depending on the client’s wants. In the lipid processing industry, fractionation is one of the steps. It is a way to separate things that are based on crystallization. The fractionation of shea butter results in two products: shea stearin and shea olein. Shea stearin is characterized by the presence of high amounts of symmetrical stearic-rich triacylglycerols such as 1,3-distearo-2-olein. Therefore, it is used in chocolate products as a cocoa butter equivalent (CBE) to improve the final product’s quality and expand its shelf life by enhancing its fat profile and stability, reducing fat migration and blooming and ensuring its softness, gloss and snap properties. Extensive application of shea olein occurs in cosmetic formulations, confectionery and specialty fat products [5]. Shea olein, with melting points ranging from 25 to 30 °C, which can be obtained depending on the fractionation condition, proved to be a rich source of bioactive compounds with a wide range of applications [4]. Oils rich in minor components (natural antioxidants) and unsaturated fatty acid play a significant function in antimicrobial activity and could enhance their antifungal activity [6]. This function was discovered to be an essential property against mycotoxigenic fungi and their dangers [7]. Whereas the effects of refining and fractionation on shea butter and the properties of its fractions are essential, more research should be conducted in this area. 

Consequently, this work aims to study the effect of the refining and fractionation processes on the characteristics of shea butter stearin and olein fractions and their mixture. Additionally, the segmentation of minor and major bioactive components between liquid and solid fractions and their effect on their antioxidant and antimicrobial activities was evaluated.

## 2. Materials and Methods

### 2.1. Materials

#### 2.1.1. Shea Butter and Its Fractions

Crude shea butter (*Vitellaria paradoxa*) and refined shea (stearin and olein) were kindly supplied by Bunge Loders Croklaan Office—Egypt, imported from Ghana and stored at 4 °C until the investigation. 

#### 2.1.2. Standards and Reagents

Standards of phytosterols, carotenoids and tocopherols, 5α-cholestane and Folin–Ciocalteu reagent were purchased from Sigma-Aldrich (Munich, Germany) and Merck (Darmstadt, Germany). A standard mixture of fatty acid methyl esters (FAME) was purchased from Supelco (Bellefonte, PA, USA). Other solvents and reagents were purchased from Sigma-Aldrich (Munich, Germany) and were of analytical (ACS) or HPLC grade.

### 2.2. Methods

#### 2.2.1. Fatty Acids Composition

The fatty acids profile was analyzed using a gas chromatography technique following the AOCS Official Method Ce 1k-07 [8]. For analysis, a gas chromatograph Trace 1300 (Thermo Scientific, Waltham, MA, USA) with a flame ionization detector (FID) was used. The used column was an SP^TM^-2560 capillary column (100 m × 0.25 mm × 0.2 µm) (Supelco, Bellefonte, PA, USA). The carrier gas was hydrogen (1.5 mL/min). For each analysis, 1 µL of the sample was injected. The oven temperature started at 160 °C for 1 min, and then increased 6 °C/min to 220 °C and it remained at this temperature for 17 min. The inlet and detector temperature were 240 °C. 

#### 2.2.2. Triacylglycerols Composition (TAG)

The TAGs were determined on the basis of Ciftci et al. [9] and Idrus et al. [10]. For TAG analysis, HPLC with ELSD detector (1260 Infinity II, Agilent Technologies, Creek Blvd Santa Clara, CA 95051 USA) was used. The temperature of the column (Infinity Lab Poroshell 120 EC-C18 4.6 × 100 mm, 2.7 Micron, Agilent Technologies, USA) was 30 °C. The initial composition of the mobile phase was: phase A acetonitrile 80%, phase B dichloromethane 20%; up to 30 min phase change to 55% A and 45% B; then, up to 40 min changing the composition of the phase to the initial one and holding it for 10 min. Samples were diluted in dichloromethane. ELSD parameters: evaporator and nebulizer temperatures 30 °C, the gas flow rate—1.60 SLM, the gain (PM)—1.0.

#### 2.2.3. Physicochemical Parameters

Crude shea butter, refined shea olein and stearin fractions and a mixture of both fractions (olein and stearin, 1:1 *w*/*w*) were analyzed to determine their physicochemical characteristics. Free fatty acids (FFA) were determined following the AOCS Official Method Ca 5a-40 [11]. The determination of unsaponifiable matter percentage (USM) was prepared according to AOCS Official Method Ca 6b-53 [12]. Total phenolics content (TPC) was measured following the procedure of Gutfinger et al. [13], which was mentioned in detail by Mohamed et al. [14]. Total flavonoid content (TFC) was determined by a modified colorimetric method described previously by Hassan et al. [15]. Peroxide value (PV) was determined according to AOCS Official Method Cd 8b-90 [16]. The procedure described by Ramadan et al. [17] was used to determine radical scavenging activity (RSA) and EC50 (the minimum effective concentration of material required to scavenge DPPH radical by 50%). Determination of slip melting point was carried out according to AOCS Official Method Cc 1-25 [18]. 

#### 2.2.4. Carotenoids Content

The shea butter samples (about 1 g) were mixed with 2 mL acetone, vortexed for 60 s and centrifuged at 10,000 rpm for 5 min. Chromatographic analysis was performed with the HPLC-DAD Agilent Technologies 1200 Rapid Resolution system (Waldbronn, Germany) equipped with a Poroshell 120, SB-C18 column (4.6 × 150 mm, 5 µm) (Agilent Technology Inc., Palo Alto, CA, USA). The mobile phase was acetonitrile and contained 0.5 g/L of triethylamine (solvent A) and methanol: ethyl acetate (55:45 *v*/*v*; solvent B) (Sigma Aldrich Chemie GmbH, Steinheim, Germany) in a gradient from 95:5 to 60:40 in 40 min, the latter proportion being maintained until the end of the run. Flow rate was set at 0.7 mL/min [19]. Detection for the UV-vis was set at 454 nm and spectrophotometric spectra of carotenoids were registered from 250 nm to 550 nm. The identification of the carotenoids was based on the comparison of UV-vis absorption wavelengths (λ_max_) and spectroscopic fine structures (% III/II). Carotenes and xanthophylls were quantified related to β-carotene and lutein (Sigma-Aldrich, Saint Louis, MO, USA), respectively. 

#### 2.2.5. Tocopherol Content

Tocopherol content was determined following Gawrysiak-Witulska et al. [20]. HPLC was used for the determination of tocopherols and the method is described in detail in Hassanein et al. [21]. A LiChrosorb Si60 column (250 × 4.6 mm; 5 µm; Merck KGaA, Darmstadt, Germany) was used.

#### 2.2.6. Phytosterols Content

The content of phytosterols was analyzed according to AOCS Official Method Ch 6-91 [22]. The shea butter samples were saponified with 1 M methanolic KOH for 18 h. Then, phytosterols were extracted with hexane/methyl tert-butyl ether (1:1, *v*/*v*). The extracts were evaporated and pyridine and BSTFA + 1% TMCS were added to dry residues. For separation, an HP 6890 series II Plus (Hewlett Packard, Palo Alto, CA, USA) was used, equipped with DB-35MS capillary column (25 m × 0.20 mm, 0.33 μm; J&W Scientific, Folsom, CA, USA). A sample of 1.0 μL was injected in splitless mode. The initial oven temperature was 100 °C for 5 min, then the temperature was programmed to 250 °C at 25 °C/min, held for 1 min, then further programmed to 290 °C at 3 °C/min and held for 20 min. The carrier gas was hydrogen (1.5 mL/min). Phytosterols were identified by comparison of retention data with standards and GC/MS (7890A/5975C VL MSD with Triple-Axis Detector, Agilent Technologies Inc., Santa Clara, CA, USA) using the same chromatographic conditions as described above.

#### 2.2.7. Oxidative Stability

A Metrohm Rancimat apparatus model 892 (Metrohm, 4800 Zofingen, Switzerland) was used to determine the induction period of shea butter samples at 110 °C ± 0.1 °C and air flow of 20 L/h following the method of Kowalski et al. [23].

#### 2.2.8. Antimicrobial Activity

The antibacterial activity of shea butter and its fractions of shea olein, shea stearin and their mixture (1:1 *w*/*w*) were determined using the well-diffusion assay [24]. In brief, Gram-positive pathogens, namely *Staphylococcus aureus*, *Bacillus cereus* and *Listeria monocytogenes* and the Gram-negative pathogens of *Escherichia coli*, *Salmonella typhi*, *Pseudomonas aeruginosa* and *Klebsiella pneumonia* were investigated for the antibacterial effect of shea butter. In disposable sterile plates for a microbiological test, tryptic soy agar media was spread for the test performance. The cork borer tool was utilized to perform the wells in the agar media in 5 mm diameters. Methanol was used to dilute the sample (1:1), and a quantity of 50 µL sample was injected into the wells. The plates were incubated (4 °C, 2 h) for rest and diffusion time, followed by microbial spreading of the bacteria to the media. The plates were incubated (37 °C, 24 h), and the inhibition zone diameter around the well was expressed in millimeter zone diameter. The more inhibition zone, the more antibacterial activity.

#### 2.2.9. Antifungal Activity

The antifungal activity of shea butter, shea olein, shea stearin and an olein stearin mixture was determined by a well diffusion assay against six toxigenic fungal strains. These toxigenic fungi were *Aspergillus parasiticus*, *A. niger*, *Penicillium notatum*, *Fusarium oxysporium*, *Rhizopus stolonifer* and *Alternaria alternate* according to the methodology described before [25]. The test fungi were inoculated onto potato dextrose agar plates using the spread plate technique, where the well formation was performed using a cork borer tool at a 5 mm diameter. The wells were then filled with 50 µL of oil. Each plate had two wells, one on each side. At the plate center, an agar disc of tested fungi was inserted individually into the dish. Each dish was analyzed after being incubated (25 °C, 96 h). The ensuing dead zones were perfectly spherical, and their lawns merged together. Complete inhibitory zones and the control well containing chloramphenicol (30 µg/mL) were measured for their diameters. The antifungal impact of shea butter components were determined as zone inhibition diameter (millimeter) against the control. The more zone diameter inhibition, the more influential the antifungal component. 

### 2.3. Statistical Analysis

The results are presented as means ± standard deviations (SD) from three replicates of each experiment. The statistical program Statistica 13.3 (TIBCO Software Inc., Palo Alto, CA, USA) was used to prepare a one-way analysis of variance (ANOVA). Probability values of less than 5% (*p* < 0.05) were considered to be significant.

## 3. Results and Discussion

The crude plant oils contain a group of active functional components such as flavonoids, phenolic compounds, tocopherols, sterols and carotenoids. The refining process is required in order to obtain odorless products with high oxidative stability. A properly conducted oil refining process should not affect the composition of fatty acids and triacylglycerols and should cause as little loss of bioactive compounds as possible. 

### 3.1. Fatty Acid Composition

Fatty acids (FAs) are involved in several activities in the body [26]. The most abundant saturated and unsaturated fatty acids (SFA, USFA) defined in crude shea butter are palmitic, stearic, oleic, arachidic and linoleic, illustrated in Table 1. In crude shea butter, the most dominant monounsaturated fatty acid (MUFA) is oleic acid (45.77%), followed by saturated stearic acid (43.53%). These results agree with Codex Alimentarius [27] for unrefined shea butter, where palmitic acid ranged from 2 to 10%, stearic acid from 25 to 50%, oleic acid from 32 to 62%, linoleic acid from 10 to 11%, linolenic acid less than 1.0% and arachidic acid less than 3.5%.

The solid fraction of refined shea stearin had the most stearic acid (60.37%) and the second most oleic acid (32.85%). Shea stearin is hard at room temperature because the ratio of stearic acid to oleic acid is high, and this makes it helpful in making bakery fat and margarine [28]. Oleic acid was the major USFA (56.03%) in refined shea olein, followed by stearic acid (30.25%). Oleic acid was higher in refined shea olein than in crude shea butter. Crude shea butter, its fractions and its mixture are good sources of omega-9 (ω-9). Linoleic acid content ranged from 2.86% to 8.06% in crude shea butter and its fractions and mixture samples. This result agrees with Codex Alimentarius [27], which reported that the linoleic acid content in crude shea butter ranged from 6 to 8%.

Compared with other oils, such as palm oil, crude shea butter has a low omega-6 (ω-6) content [29]. Otherwise, in crude shea butter, linolenic acid (18:3, ω-3) was found only in a small percentage (0.13%). Concerning the mixture between refined shea stearin and olein fractions (1:1 *w*/*w*), it was found that the SFA was nearly equal to the USFA (50.84% and 49.16%, respectively). In addition, the presence of (20:1) fatty acid was detected in a small ratio in the mixture, although it was not found in shea stearin. 

As shown in Table 1, crude shea butter had 48.17% SFAs, 46.05% MUFAs and 5.98% PUFAs. These results agree with Goumbri et al. [30], who reported that SFA in crude shea butter ranged from 3% to 58%, MUFA from 3% to 68% and PUFA from 1% to 11%.

It was noticed that the total value of SFA was lower than that of USFA in the case of crude shea butter and refined shea olein. On the other hand, the total amount of SFA in refined shea stearin was higher than the total amount of USFA. About the mixture, it was seen that the total SFA was almost the same as the total USFA.

### 3.2. Triacylglycerol Composition

In terms of the triacylglycerol (TAG) composition of crude shea butter, nine TAGs with different critical pairs ranging from ECN 46 to ECN 50 were separated (Table 2). The main TAGs were SOS (46.77%) and SOO (42.12%) in crude shea butter. These results agree with Maranz et al. [31], who found that the main TAGs in shea butter were SOS and SOO. Additionally, these results agree with those of Honfo et al. (2014), who reported that SOS ranged from 13% of total TAGs in Ugandan shea butter to 45% in shea butter from Burkina Faso, while SOO was the highest (28–30%) in Ugandan and Malian shea butter. 

After refining and fractionation, only four TAGs were detected in shea stearin (Table 2), where the major TAG was SOS (90.71%). On the other hand, shea olein contained nine TAGs, the most common of which was SOO (72.66%), followed by SLO (8.45%) and OOO (7.23%). Other TAGs, such as OLO, SLL, POO, POS and AOO, are found in crude shea butter, shea olein and the mixture, but in smaller amounts. Other TAGs, such as PPS, were present only in shea stearin (1.59%). The mixture’s TAGs were similar to those of crude shea butter, where SOS and SOO were the principal TAGs (32.39% and 57.76%, respectively).

### 3.3. Free Fatty Acids

Free fatty acids (FFAs) of shea butter is a measure of the extent to which the triacylglycerols in the butter have been broken down by lipase or other things such as heat and light. It is often used as a general indicator of the condition and edibility of the oil. FFAs of crude shea butter were found to be 12.2% (Table 3). This result agrees with Goumbri et al. [30], who reported that FFAs in crude shea butter ranged from 9.0% to 23.9%. The refining process before fractionation caused removed FFAs, and they were present in a negligible amount in the individual as well as mixture of olein and stearin (Table 3). Womeni et al. [32] and Nkouam et al. [33] reported that the FFAs of shea butter vary from 0.0% to 10.6%, with an average of 4.1%. However, Nkouam et al. [33] found high FFAs (64.1%) in shea butter extracted by supercritical CO_2_ in kernels that had been stored for two years. The FFA content of shea butter is affected by the duration of storage, processing, packaging material, germinating stage of the fruit of the shea nut and general climatic conditions [34]. This could explain why the values reported here differ from other studies.

### 3.4. Unsaponifiable Matter

Unsaponifiable matter (USM) refers to substances dissolved in fat that are insoluble in an aqueous solution but are soluble in organic solvents after saponification. It is usually composed of sterols, fatty alcohols, tocopherols, triterpene alcohols and hydrocarbon (squalene) in almost all vegetable fats and oils [35]. The results found that the percentage of USM in crude shea butter was 5.1% (Table 3). After refining and fractionation, USM was 0.2%, 6.4% and 4.1% in refined shea stearin, refined shea olein and in the mixture, respectively. The USM of crude shea butter was reported to be relatively higher than that of other vegetable oils, at up to 4% USM [36]. Our results did not agree with those of Goumbri et al. [30], who reported that the percentage of USM in crude shea butter ranged from 7.00% to 7.61%. These differences may be due to region, climate and extraction method. Our results agree with Codex Alimentarius [27], which found that the USM ranged from 1% to 19%.

It was noticed that the increase in USM after fractionation in shea olein may be due to the soluble selectivity of bioactive components in the liquid part and, at the same time, because shea olein is a part of crude shea butter; this led to an increase in the percentage of USM in shea olein [36].

It was reported by Nahm [37] that the percentage of USM in shea nut oil was 0.95%; this was significantly (*p* < 0.05) higher than that given by the crude palm oil of 0.55%. The high USM content of shea butter shows that the oil is rich in desirable bioactive components such as antioxidants, antimicrobials, anti-inflammatory substances and fat-soluble vitamins. It has also been used to lower cholesterol levels by a pharmaceutical company, BSP Pharma [38].

### 3.5. Total Phenolic Content

Our results found that crude shea butter’s total phenolic content (TPC) was 0.211 mg/g (Table 3). This agrees with those of Goumbri et al. [30], who reported that the TPC of crude shea butter ranged from 0.0 to 4.0 mg/g.

The refining process did not cause a statistically significant decrease in these compounds. The content of TPC in both refined shea stearin and olein fractions was 0.19 mg/g. In the case of the mixture, it was 0.21 mg/g.

Zacchi and Eggers [39] showed that the pretreatment of seeds had an influence on the content of phenolic compounds in crude oil. Their content was slightly reduced during degumming, but neutralization resulted in a complete removal of these compounds. During the physical refining, the amount and type of bleaching clay affected the content of polyphenols.

### 3.6. Total Flavonoid Content

Concerning the subgroup constituents of polyphenols, namely, flavonoids (potentially powerful antioxidants), Table 3 shows that a higher total flavonoid content (TFC) was found in crude shea butter (40.14 mg catechin/100 g oil). This value decreased more than 34% for stearin and olein fractions after refining and fractionation, and after blending them the content of flavonoids was 29.2 mg catechin/100 g of oil.

The refining process of rice bran oil caused about a 75% decrease in TFC [40]. Since the oil refining process consists of several stages, it would be necessary to adapt the conditions of each stage to the individual oil, its composition and oxidative stability.

### 3.7. Peroxide Value

The peroxide value (PV) of crude shea butter was 5.91 meq O_2_/kg; this value was significantly (*p* < 0.05) higher than that of the refined shea olein at 0.63 meq O_2_/kg (Table 3). The PV of refined shea stearin was 0.24 meq O_2_/kg. These results agree with Njoku et al. [41] and Dandjouma et al. [42], who reported that PV ranges from 0.5 meq O_2_/kg to 29.5 meq O_2_/kg for crude shea butter. PVs in this work for shea butter samples were lower than 10 meq O_2_/kg, which agrees with Codex Alimentarius [27]. Ghohestani [43] reported that fresh oil usually has PVs below 10 meq O_2_/kg, but if this value increases to between 20 and 40 meq O_2_/kg, complex oil changes [44] explain that shea butter must have a PV of less than 10 meq O_2_/kg for use in food applications.

### 3.8. Oxidative Stability Parameters

From Table 3, it was found that crude shea butter and refined shea olein have nearly the same RSA (28.33% and 28.61%, respectively) and are more stable than that of shea stearin (6.79%) and the mixture (17.68%). 

From the results obtained for EC_50_, it is clear that crude shea butter and refined shea olein (189.04 mg/mL and 183.31 mg/mL, respectively) show more stability than that of refined shea stearin (835.83 mg/mL). In addition, the mixture showed more stability than the refined shea stearin but was less stable than the crude shea butter and shea olein. These results could be supported by the results of USM % (Table 3), which was found to be higher in both crude shea butter samples and olein fractions. As previously mentioned, USM consists of all the bioactive compounds that protect the oil and increase its stability [36].

### 3.9. Melting Point

The melting point (MP) is a critical aspect of the traditional processing of shea butter [45]. From the results in Table 1, it was noticed that the MP of crude shea butter was 32 °C. After fractionation, the MPs of shea stearin and olein were 36 and 25 °C, respectively. The MP of the mixture was 31 °C. These results agree with those of Womeni et al. [32], who reported that MP’s vary between 25 °C and 45 °C, with an average of 35.9 °C, depending on shea origin and processing method. It is stated that an MP’s closeness to body temperature is an attribute that makes it suitable as a base for many purposes, especially medicines. Depending on the fatty acid profile of the samples, these findings could be anticipated (Table 1). Shea stearin, which has the greatest SFA percentage (64.30%), also has the highest melting point, whereas shea olein, which has the lowest SFA percentage, has the lowest melting point.

### 3.10. Total Carotenoid Content

The content of carotenoids was determined in all samples by HPLC method, but they were not detected. Mbah et al. [46] analyzed the pigments from shea butter and their adsorption on acid-activated Cameroonian local clays. They determined the content of total pigments using a spectrophotometric method at 295 nm. This wavelength is typical for polyphenols, but not for carotenoids and chlorophyll. 

### 3.11. Tocopherols

Crude shea butter contains alpha (α), beta (β), gamma (γ) and delta (δ)-tocopherols (T), as shown in Table 4. α-T was the major tocopherol in crude shea butter, accounting for about 84.70% of the total tocopherols, followed by β-T (10.34%). γ- and δ-T were found in smaller amounts (3.60% and 1.50%, respectively). These results agree with Davrieux et al. [2] and Allal et al. [47], who reported that α-T was the largely predominant form, amounting to 64%, while β- and δ-T were less than 1.5 µg/g in crude shea butter.

It was noticeable that γ- and δ-T were wholly lost after the refining of shea butter samples (shea olein, shea stearin and the mixture) (Table 4).

The results in Table 4 found that the highest content of total T was in crude shea butter and refined shea olein samples, which amounted to 10.06 mg/100g and 9.25 mg/100 g, respectively. It was, however, significantly lower in shea stearin (1.27 mg/100 g). It was also clear that the mix of shea olein and stearin fractions has a moderate amount of total tocopherols. The loss of total tocopherol in the shea stearin fraction was 87.4% and 8.0% in the case of the shea olein fraction compared with crude shea butter. These results agree with those of Wong et al. [48], who reported that no tocopherols were lost after fractionation. However, the tocopherols were concentrated in the olein fraction and depleted in the stearin. Tocopherols are phenolic antioxidants that stop lipids from turning rancid by eliminating free radicals and interacting with singlet oxygen. Alpha-tocopherol blocks the effects of singlet oxygen during sensitive photo oxidation in vegetable oils [49].

### 3.12. Phytosterol Composition

It was clear that shea butter’s phytosterols were made up of parts that are not found in most vegetable oils. The most abundant triterpenes (squalene, lanosterol, α-amyrin and β-amyrin) were found in shea butter samples (Table 5). Squalene is a natural triterpene that is an important step in the production of phytosterol, and it was found in crude shea butter in higher amounts (0.20 mg/g). After refining and fractionation, it was noticeable that the squalene content decreased in refined shea stearin and olein, as well as in the mixture, to 0.03 mg/g, 0.05 mg/g and 0.04 mg/g, respectively. According to Table 5, the squalene loss percentage increased from 76.9% to 87.2% after the refining and fractionation processes.

Lanosterol serves as a precursor of phytosterols and various steroidal hormones. It is found in higher amounts in crude shea butter oil (0.25 mg/g). After refining, it decreased to 0.04 mg/g, 0.24 mg/g and 0.18 mg/g in shea stearin, shea olein and the mixture, respectively. After the refining and fractionation processes, the loss % in lanosterol was 84%, 4% and 28% in shea stearin, shea olein and the mixture, respectively.

From the results, it was noticed that α- and β-amyrin were higher in crude shea butter (13.09 mg/g and 7.32 mg/g). These findings are consistent with Di Vincenzo et al. [50], who discovered that the main terpenic compounds are the triterpenes alcohol α- and β-amyrin. Refining and fractionation showed no change in an amount of α- and β-amyrin in the refined shea olein (12.43 mg/g and 7.12 mg/g, respectively) compared with crude shea butter. It highly decreased in shea stearin (2.32 mg/g and 1.31 mg/g, respectively). In addition, it was found in a reasonable amount in the mixture of shea olein and stearin (1:1 *w*/*w*) (9.31 mg/g and 5.28 mg/g, respectively). The loss % of α- and β-amyrin were 82.3% and 82.1%, respectively, for shea stearin, while it was 5.0% and 2.7% in the case of shea olein and 28.9% and 27.9%, respectively, in the case of the mixture.

The results showed that minor bioactive components (tocopherols and sterols) selectively preferred to migrate to the liquid fraction (olein).

### 3.13. Induction Period (IP)

The susceptibilities of the crude shea butter, refined shea stearin, refined shea olein and their mixture to oxidation were evaluated by the Rancimat method. IP can determine the Rancimat test’s endpoint to the oxidation curve’s inflection point [51]. The IP length is considered a relative measure of the stability of oils. From the results, it was found that the mixture (olein and stearin at 1:1 *w*/*w*) had the highest IP (18.5 h), followed by shea olein (15.9 h). The high stability of the mixture may be due to the synergistic effect between minor components and the balance between SFA and USFA. In addition, it is free from FFA and peroxide components. The results in Figure 1 also show that the IP of the crude shea butter was the lowest IP (11.5 h), while shea stearin had a moderate IP (12.5 h). Additionally, although shea olein contains the highest PUFA (8.06%), it showed high IP, which may be due to its high content of minor bioactive components that prefer to migrate into it as the liquid fraction. Apparently, the types and quantities of tocopherol, sterol and terpenoid in the shea samples seem to be adequate to safeguard its unsaturated fatty acids from oxidation.

### 3.14. Antimicrobial Activity of Shea Butter Samples

#### 3.14.1. Antibacterial Activity

The results recorded for the antibacterial activity impact of shea butter and its fractions were estimated against two types of pathogens (Gram-positive and -negative bacteria), reflecting high activity (Figure 2). The refined shea olein was recorded as the most effective component, followed by the crude shea butter. In contrast, the impact of the refined mix of shea stearin and olein fractions (1:1 *w*/*w*) was the lowest and came last. These results could be explained through the evaluated parameters of each fraction and its antioxidant activity and minor bioactive component content. The refined olein and crude shea butter samples recorded higher contents of the minor bioactive components, illustrating their antibacterial activity. However, most of these minor bioactive components are transferred to the refined olein fraction during the fractionation process, which explains this fraction’s high antibacterial activity. The antibacterial activity is linked to minor bioactive components such as tocopherols [52], phenolic acids and flavonoids [53] and sterols [54]. The high efficiency of the shea olein fraction could also be linked to its high content of unsaponifiable matter, as reported by Ornella et al. [55].

Different shea butter fractions appeared to have a more significant effect on Gram-positive bacteria than Gram-negative bacteria. These differences were significant in most cases when testing against pathogenic Gram-negative bacteria. Pathogenic Gram-negative bacterial cell wall components’ sensitivity to shea-microcomponents could explain this effect. The reconstituted mixture of shea butter fractions (stearin:olein: 1:1 *w*/*w*) was the least effective treatment for Gram-positive and Gram-negative bacteria [56]. This could be because some crucial parts were lost during the stages of refining and fractionation [57]. Additionally, the interaction between the ingredients in the natural state of shea butter may have a more synergistic effect between the active ingredients than in the case of their recombination [58].

In this regard, the results reflected a distinguished impact of shea olein refined fraction due to its richness of active components against microbial contamination. The results pointed out the high content of total T for refined olein, which was recorded closely to the crude. Again, the total content of phytosterols behave such as tocopherol (close for crude and refined olein). However, for the crude shea, these bioactive components provide the defense of the two parts (olein and stearin) against microbial contamination, which may affect the efficacy of these components. The concentrated effect of these bioactive components (TPC, TFC, T and phytosterols), as well as their synergism or interaction impact, could present an explanation for the higher activity of the olein fraction [59]. The refining process of vegetable oil concentrated some components to such fractions [60]. This action may illustrate the higher activity of shea olein fraction as an antimicrobial agent. Additionally, the shelf life of shea olein was shown by a higher induction period value compared with the crude one and shea stearin (Figure 1).

#### 3.14.2. Antifungal Activity

The antifungal efficacy of crude shea butter, refined fractions (stearin and olein) and their mixture (stearin:olein (1:1 *w*/*w*)) was investigated, and the results are shown in Figure 3. The findings show that crude shea butter significantly impacted four different types of toxic fungi. Refined shea stearin came in second place in antifungal action. There was still an antifungal effect from the shea olein fraction, which was quite similar to the effect from the shea stearin fraction. The mixture of shea olein and stearin fractions showed the lowest antifungal activity. Several bioactive components in crude shea butter may explain its high antifungal activity. In addition to phenolic compounds, it is abundant in unsaturated triglycerides, phytosterols and tocopherol. A role in the antifungal activity of black cumin and Roselle oil was found by Badr et al. [25], and this role might be related to the concentration of minor components such as γ-tocopherol. Losses in minor and bioactive components may account for the decline in antifungal properties of refined shea relative to the combination of shea fractions (stearin and olein).

The impact of crude shea was as the most effective antifungal agent due to its higher content of total T, TPC, TFC and sterols. The previous research reported a distinguished function for these components, particularly for the T content [61] and phytosterols [62]. The impact of these components includes cell wall changes and molecular changes impact [63]. The result of antifungal inhibition was also valuable for the shea stearin fraction. This fraction was distinguished for its content of palmitic acid and the SFA. For the triglycerides, it also possessed a high content of the SOS. These bioactive components were reported to have antifungal activity [64], particularly against toxigenic fungal strains [65].

## 4. Conclusions

It was noticeable that the sterol composition of shea butter contained unique components not found in most vegetable oils, namely, squalene, lanosterol, α-amyrin and β-amyrin. The refining process markedly affected minor components, decreasing total tocopherols, sterols and their fractions. It was noticed that USM was markedly lower after refining in the case of the shea stearin fraction. On the contrary, it was increased in the case of the shea olein fraction. In the mixture of olein and stearin (1:1 *w*/*w*), sample USM decreased little. After refining, it was noticeable that γ-and δ- tocopherols were completely lost in shea olein and stearin fractions. Total tocopherols and α-tocopherol were found in shea olein nearly equal to that in crude shea butter. The refining process caused a decrease in TPC and TCC in both fractions and their mixture. In contrast, the fractionation process affected the distribution of minor components between the liquid fraction (olein) and a solid fraction (stearin). 

It was noticed that after refining and fractionation, the mixture of stearin and olein fractions (1:1 *w*/*w*) had a higher IP than both individual fractions. The shea olein fraction showed a higher IP than the shea stearin fraction and crude shea butter. Consequently, the distribution of the minor components affected the antioxidant, antimicrobial and antifungal activities of shea butter and its fractions. Crude shea butter showed distinguished antifungal activity due to the combination of various minor components possessing antioxidant potency. The shea olein fraction showed antimicrobial efficiency due to its high antioxidant and bioactive component content.

## Figures and Tables

**Figure 1 foods-12-01626-f001:**
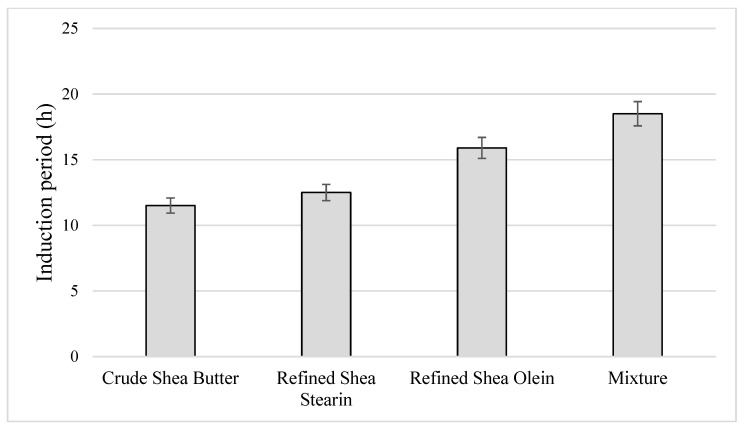
Induction period (h) of crude shea butter, refined shea stearin, refined shea olein and a mixture of olein:stearin 1:1 *w*/*w*.

**Figure 2 foods-12-01626-f002:**
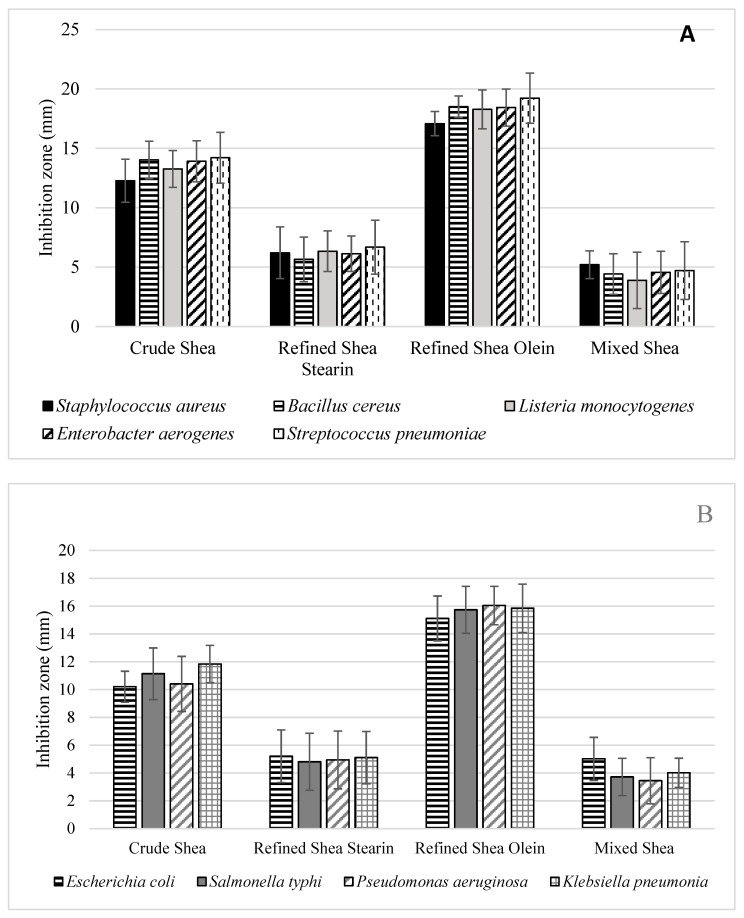
Antibacterial activity of crude shea butter, refined shea stearin, refined shea olein and a mixture (olein:stearin 1:1 *w*/*w*) determined against Gram-positive (**A**) and Gram-negative (**B**) pathogens.

**Figure 3 foods-12-01626-f003:**
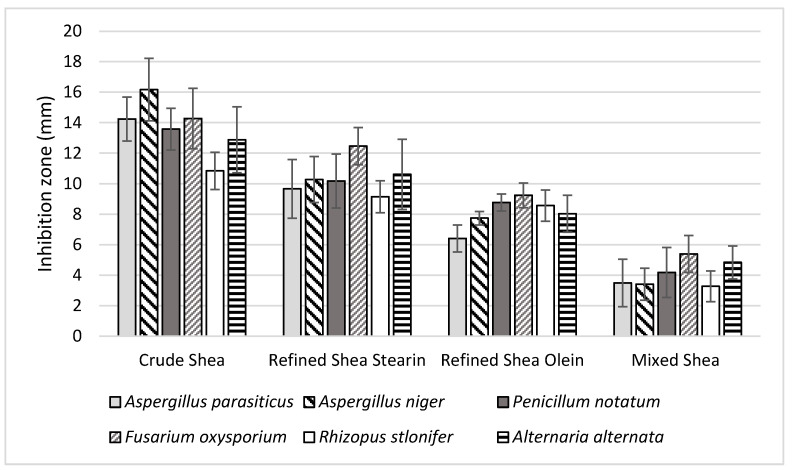
Antifungal activity of crude shea butter, refined shea stearin, refined shea olein and a mixture of olein:stearin (1:1 *w*/*w*) determined against toxigenic fungal strains. The results are expressed as mean percentages ± SD; (*n* = 3; *p* ˂ 0.05).

**Table 1 foods-12-01626-t001:** The fatty acid composition (%) of crude shea butter, refined shea stearin, refined shea olein and a mixture of olein:stearin (1:1 *w*/*w*).

Fatty Acids (%)	Crude Shea Butter	Refined Shea Stearin	Refined Shea Olein	Mixture
C16:0	3.17 ± 0.02 ^b^	2.43 ± 0.01 ^d^	3.89 ± 0.01 ^a^	3.06 ± 0.02 ^c^
C18:0	43.53 ± 0.07 ^c^	60.37 ± 0.16 ^a^	30.25 ± 0.00 ^d^	46.37 ± 0.08 ^b^
C18:1	45.77 ± 0.09 ^b^	32.85 ± 0.08 ^d^	56.04 ± 0.01 ^a^	43.79 ± 0.07 ^c^
C18:2 n-6	5.85 ± 0.02 ^b^	2.86 ± 0.00 ^d^	8.06 ± 0.01 ^a^	5.13 ± 0.02 ^c^
C18:3 n-3	0.13 ± 0.00 ^b^	ND ^b^	ND ^b^	ND ^b^
C20:0	1.47 ± 0.02 ^ab^	1.50 ± 0.06 ^a^	1.26 ± 0.01 ^c^	1.42 ± 0.01 ^b^
C20:1	0.28 ± 0.02 ^b^	ND ^c^	0.52 ± 0.01 ^a^	0.25 ± 0.02 ^b^
SFA	48.17 ± 0.02 ^c^	64.30 ± 0.14 ^a^	35.40 ± 0.07 ^d^	50.84 ± 0.01 ^b^
MUFA	46.05 ± 0.02 ^b^	32.85 ± 0.01 ^d^	56.55 ± 0.01 ^a^	44.05 ± 0.01 ^c^
PUFA	5.98 ± 0.01 ^b^	2.85 ± 0.01 ^d^	8.06 ± 0.01 ^a^	5.13 ± 0.01 ^c^

Where 16:0—palmitic acid, 18:0—stearic acid, 18:1—oleic acid, 18:2—linoleic acid, 18:3—linolenic acid, 20:0—arachidic acid and 20:1—gadoleic acid; SFA—saturated fatty acids, MUFA—monounsaturated fatty acids, PUFA—polyunsaturated fatty acids, ND—not detected. ^a–d^—Different symbols within the same rows indicate significant difference at *p* < 0.05.

**Table 2 foods-12-01626-t002:** Triacylglycerol composition (%) of crude shea butter, refined shea stearin, refined shea olein and a mixture of olein:stearin (1:1 *w*/*w*).

ECN	Triacylglycerols (%)	Crude Shea Butter	Refined Shea Stearin	Refined Shea Olein	Mixture
46	OLO	0.42 ± 0.12 ^b^	ND ^c^	1.06 ± 0.37 ^a^	0.45 ± 0.09 ^b^
46	SLL	0.39 ± 0.13 ^b^	ND ^c^	0.96 ± 0.19 ^a^	0.39 ± 0.11 ^b^
48	OOO	2.89 ± 0.30 ^b^	ND ^d^	7.24 ± 0.51 ^a^	2.28 ± 0.03 ^c^
48	SLO	4.23 ± 0.19 ^b^	ND ^d^	8.45 ± 0.26 ^a^	2.49 ± 0.15 ^c^
48	POO	0.72 ± 0.08 ^c^	ND ^d^	2.59 ± 0.11 ^a^	1.11 ± 0.15 ^b^
50	SOO	42.12 ± 0.42 ^b^	4.75 ± 0.61 ^d^	72.66 ± 3.03 ^a^	32.39 ± 0.80 ^c^
50	AOO	0.32 ± 0.04 ^b^	ND ^c^	0.68 ± 0.10 ^a^	0.31 ± 0.12 ^b^
50	PPS	ND ^b^	1.59 ± 0.05 ^a^	ND ^b^	ND ^b^
50	POS	2.15 ± 0.26 ^b^	2.94 ± 0.45 ^a^	2.01 ± 0.25 ^b^	2.82 ± 0.07 ^a^
50	SOS	46.77 ± 0.69 ^c^	90.72 ± 1.12 ^a^	2.82 ± 0.07 ^a^	57.77 ± 1.02 ^b^

Where ECN—equivalent carbon number (carbon number of TAG—2 × number of double bonds), P—palmitic, S—stearic, O—oleic, L—linoleic, A—arachidic, ND—not detected. ^a–d^—Different symbols within the same row indicate significant difference at *p* < 0.05.

**Table 3 foods-12-01626-t003:** Physicochemical characteristics and oxidative stability parameters of crude shea butter, refined shea stearin, refined shea olein and a mixture of olein:stearin (1:1 *w*/*w*).

Parameters	Crude Shea Butter	Refined Shea Stearin	Refined Shea Olein	Mixture
FFAs (%)	12.24 ± 0.30 ^a^	0.00 ± 0.00 ^b^	0.00 ± 0.00 ^b^	0.11 ± 0.00 ^b^
USM (%)	5.06 ± 0.11 ^b^	0.22 ± 0.04 ^d^	6.38 ± 0.57 ^a^	4.06 ± 0.11 ^c^
TPC (mg/g)	0.21 ± 0.01 ^a^	0.19 ± 0.01 ^b^	0.19 ± 0.01 ^b^	0.21 ± 0.02 ^a^
TFC (mg catechin/100 g)	40.14 ± 0.20 ^a^	26.46 ± 0.46 ^b^	27.61 ± 0.70 ^c^	29.20 ± 0.44 ^d^
PV (meq O_2_/kg)	5.91 ± 0.40 ^a^	0.24 ± 0.08 ^b^	0.63 ± 0.01 ^c^	0.30 ± 0.04 ^b^
RSA (%)	28.33 ± 0.12 ^b^	6.79 ± 0.20 ^d^	28.61 ± 0.13 ^a^	17.68 ± 0.15 ^c^
EC_50_ (mg/mL)	189.04 ^c^	835.83 ^a^	183.31 ^c^	315.86 ^b^
Melting point (°C)	32 ± 0 ^b^	36 ± 0 ^a^	25 ± 0 ^d^	31 ± 1 ^c^

Where FFAs—free fatty acids, TPC—total phenolic content, TFC—total flavonoid content, PV—peroxide value, RSA—radical scavenging activity, EC_50_—the minimum effective concentration of material required to scavenge DPPH radical by 50%. ^a–d^—Different symbols within the same rows indicate significant difference at *p* < 0.05.

**Table 4 foods-12-01626-t004:** Tocopherol content (mg/100 g) of crude shea butter, refined shea stearin, refined shea olein and a mixture of olein:stearin (1:1 *w*/*w*).

Tocopherols (mg/100 g)	Crude Shea Butter	Refined Shea Stearin	Refined Shea Olein	Mixture
α-T	8.52 ± 0.03 ^a^	1.06 ± 0.03 ^d^	8.28 ± 0.03 ^b^	4.58 ± 0.08 ^c^
β-T	1.04 ± 0.02 ^a^	0.21 ± 0.04 ^d^	0.97 ± 0.05 ^b^	0.36 ± 0.03 ^c^
γ-T	0.36 ± 0.02 ^a^	ND ^b^	ND ^b^	ND ^b^
δ-T	0.15 ± 0.03 ^a^	ND ^b^	ND ^b^	ND ^b^
Total	10.06 ± 0.06 ^a^	1.27 ± 0.07 ^d^	9.25 ± 0.02 ^b^	4.94 ± 0.11 ^c^

Where α-T—α-tocopherol, β-T—β-tocopherol, γ-T—γ-tocopherol, δ-T—δ-tocopherol, ND—not detected. ^a–d^—Different symbols within the same column indicate significant difference at *p* < 0.05.

**Table 5 foods-12-01626-t005:** Phytosterol content (mg/g) of crude shea butter, refined shea stearin, refined shea olein and a mixture of olein:stearin (1:1 *w*/*w*).

Sterols (mg/g)	Crude Shea Butter	Refined Shea Stearin	Refined Shea Olein	Mixture
Squalene	0.20 ± 0.01 ^a^	0.03 ± 0.00 ^d^	0.05 ± 0.00 ^b^	0.04 ± 0.00 ^c^
Lanosterol	0.25 ± 0.01 ^a^	0.04 ± 0.00 ^c^	0.24 ± 0.00 ^a^	0.18 ± 0.01 ^b^
α-Amyrin	13.09 ± 1.33 ^a^	2.32 ± 0.03 ^c^	12.43 ± 0.59 ^a^	9.31 ± 0.23 ^b^
β-Amyrin	7.32 ± 0.75 ^a^	1.31 ± 0.01 ^c^	7.12 ± 0.39 ^a^	5.28 ± 0.14 ^b^
Total	20.83 ± 0.62 ^c^	3.69± 0.03 ^a^	19.85 ± 0.51 ^c^	14.81 ± 0.21 ^b^

^a–d^—Different symbols within the same row indicate significant difference at *p* < 0.05.

## Data Availability

The data presented in this study are available on request from the corresponding author.

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
