# Peer review of "Effect of Refining and Fractionation Processes on Minor Components, Fatty Acids, Antioxidant and Antimicrobial Activities of Shea Butter"

_foods, 2023, doi:10.3390/foods12081626_

Round 1

Reviewer 1 Report

It is interesting to explore the fatty acid compositions, lipid compositions, minor bioactive ingredients, and the involved bioactivities of the fractions of shea butter. Some  comment comments are listed below:

Why did the authors investigate total phenolic and flavonoid content? Have the authors analyzed the compositions by HPLC or HPLC-MS? 

It is suggested to discuss more about the relationship between the compositions and the properties, such as oxidative stability, antioxidant, and antimicrobial.

Author Response

Reviewer 1: It is interesting to explore the fatty acid compositions, lipid compositions, minor bioactive ingredients, and the involved bioactivities of the fractions of shea butter. Some comments are listed below:

Point 1: Why did the authors investigate total phenolic and flavonoid content? Have the authors analyzed the compositions by HPLC or HPLC-MS? 

Response 1:

This article aimed to study the effect of refining and fractionation processes on the minor components, fatty acids, antioxidants, and antimicrobial activities of shea butter. Therefore, the focus was on studying the influence of minor components, especially phytosterols and tocopherols, which chromatographic analyzed. Concerning phenols and flavonoids, whose presence is limited in shea butter, in addition to the loss of a high percentage of them during the refining processes, from our view, it was sufficient to study their total content in the samples.

Point 2: It is suggested to discuss more about the relationship between the compositions and the properties, such as oxidative stability, antioxidant, and antimicrobial.

Response 2:

Based on your valuable opinion, many sentences and discussions have been added to the manuscript on the relationship between formulations and properties, such as oxidative stability, antioxidants, and antimicrobials (writed in red color).

Reviewer 2 Report

The various processes will bring different influence on the quality and component composition of food products. In this article submitted by Adel G. Abdel-Razek et al., and entitled "Effect of refining and fractionation processes on minor components, fatty acids, antioxidant and antimicrobial activities of shea butter", the author investigated the effect of refining and fractionation processes on components composition, as well as the antioxidant and antimicrobial activity. However, the experimental design and data analysis of this article were simple and lack of depth; Besides, several questions should be addressed.

1. As we known that oil refining process includes multiple processes, in this article, the author should provide the refining and fractionation processes of shea butter in detail.

2. In the section of “2.2.8 Antimicrobial activity”, the author described that the sample was diluted with methanol (1:1), and injected into the wells after bacteria being inoculated to the media; After that, the plates were incubated at 37 ℃ for 24h. However, the methanol is prone to volatilize, the plates should be put at 4 ℃ for 2 h before incubation at 37 ℃. Additionally, the diameter of the cork borer tool mentioned in this section should be provided.

3. In this article, the author only simply described the data, and the deep analysis of the data was not conducted. The discussion was also lack of depth.  

4. In line of 29, “The antibacterial activity was higher, but the antifungal activity was lower than in crude shea butter.” should replaced with “The antibacterial activity was higher, but the antifungal activity was lower than crude shea butter.”

Author Response

Reviewer 2: The various processes will influence the quality and component composition of food products differently. In this article submitted by Adel G. Abdel-Razek et al., entitled "Effect of refining and fractionation processes on minor components, fatty acids, antioxidant and antimicrobial activities of shea butter", the author investigated the effect of refining and fractionation processes on components composition, as well as the antioxidant and antimicrobial activity. However, this article's experimental design and data analysis was simple and lacked depth; Besides, several questions should be addressed.

Point 1: As we know, the oil refining process includes multiple processes; in this article, the author should provide the refining and fractionation processes of shea butter in detail.

Please add some clarifications:

Response 1:

The added scentences to clarify the refining process in lines from 53 to 59.

It is well known that shea butter refining involves various steps. When making refined shea butter from raw shea butter, there are four important steps: de-gumming, neutralization, bleaching, and deodorization. The ingredients that aren't good for consumption are removed from the butter at each stage of the refining process. Also, the refining process unintentionally removes some desirable natural bioactive components. After being refined, shea butter can be fractionated into shea butter olein and stearin, depending on the client's wants.

Point 2: In the section "2.2.8 Antimicrobial activity," the author stated that the sample was diluted with methanol (1:1) and injected into the wells after the bacteria was inoculated to the media; the plates were then incubated at 37 °C for 24 hours. However, the methanol is prone to volatilize; the plates should be put at 4 °C for 2 h before incubation at 37 °C. Also, the diameter of the cork borer tool should be mentioned in this section.

Response 2: Thanks for your valuable feedback, This point was correctly added to the methodology. The missing word was regarded as a typographical mistake.

Regarding the cork borer diameter, it was already highlighted in yellow.

The correct sentence will be:

 ”The antibacterial activity of shea butter and its fractions of shea olein, shea stearin, and their mixture (1:1 w/w) were determined using the well-diffusion assay [24]. In brief, Gram-positive pathogens, namely Staphylococcus aureus, Bacillus cereus, Listeria monocytogenes, Gram-negative pathogens of Escherichia coli, Salmonella typhi, Pseudomonas aeruginosa, and Klebsiella pneumonia were investigated for the antibacterial effect of shea butter. In disposable sterile plates for a microbiological test, tryptic soy agar media was spread for the test performance. The cork borer tool was utilized to perform the wells in the agar media in 5 mm diameters. Methanol was used to dilute the sample (1:1), and a quantity of 50µL sample was injected into the wells. The plates were incubated (4 ºC, 2h) for rest and diffusion time, followed by microbial spreading of the bacteria to the media. The plates were incubated (37ºC, 24h), and the inhibition zone diameter around the well was expressed in millimeter zone diameter. The more inhibition zone, the more antibacterial activity.

Point 3: In this article, the author only described the data, and a deep analysis of the data was not conducted. The discussion was also lack of depth. 

Response 3 : Many phrases and discussions have been added to the manuscript in response to your valuable comments (writed in red color).

Point 4:  In line 29, “The antibacterial activity was higher, but the antifungal activity was lower than in crude shea butter.” should replaced with “The antibacterial activity was higher, but the antifungal activity was lower than crude shea butter.”

Response 4: Thanks for your correction; this point was done as correctly.

New sentence is :

" The antibacterial activity was higher, but the antifungal activity was lower than crude shea butter."